# A Flower Bud from the Lower Cretaceous of China

**DOI:** 10.3390/biology11111598

**Published:** 2022-11-01

**Authors:** Li-Jun Chen, Xin Wang

**Affiliations:** 1Key Laboratory of National Forestry and Grassland Administration for Orchid Conservation and Utilization, the National Orchid Conservation Center of China and the Orchid Conservation & Research Center of Shenzhen, Shenzhen 518114, China; 2State Key Laboratory of Palaeobiology and Stratigraphy, Nanjing Institute of Geology and Palaeontology and CAS Center for Excellence in Life and Paleoenvironment, Chinese Academy of Sciences, 39 East Beijing Road, Nanjing 210008, China

**Keywords:** fossil, Lower Cretaceous, China, angiosperms, flower, bud

## Abstract

**Simple Summary:**

Flowers are beautiful due to their perianth which is frequently colourful and conspicuous. The existence of the colourful perianth is closely related to insect pollination in extant angiosperms. The Early Cretaceous (1.25 million years ago) Yixian Formation in Northeastern China is famous for its great diversity of reproductive organs of early angiosperms. However, unlike typical flowers in extant angiosperms, the previously documented fossil flowers are “naked”, namely, they do not have typical perianth, suggestive of a strategy different than the extant one adopted in the reproduction of early angiosperms. However, without fossil evidence, whether there is perianth and whether androecium and gynoecium are protected in early flowers are open questions. In this paper, we document the first flower bud fossil, *Archaebuda lingyuanensis*, from the Yixian Formation. Besides being the first recorded flower bud (which is fragile and unlikely to be fossilized) in the Early Cretaceous, the flower bud does protect its internal parts, which are vulnerable to various attacks and harm, using the perianth. In addition, perianth plays an important role in attracting insects for successful pollination of early angiosperms. This knowledge on the reproduction of early angiosperms is otherwise unavailable if only the previous fossils are taken into consideration.

**Abstract:**

Background: Although various angiosperms (including their flowers) have been reported from the Yixian Formation (Lower Cretaceous) of China, which is famous worldwide for its fossils of early angiosperms, no flower bud has hitherto been seen in the Early Cretaceous. Such a lack of examples hinders our understanding of the evolution of flowers. Methods: The specimen studied in the present paper was collected from an outcrop of the Yixian Formation (the Barremian-Aptian, Lower Cretaceous) of Dawangzhangzi in Lingyuan, Liaoning, China. The specimen was photographed using a Nikon D200 digital camera, its details were observed and photographed using a Nikon SMZ1500 stereomicroscope, and some of its details were observed using a Leo 1530 VP SEM. Results: We report a fossilized flower bud, *Archaebuda lingyuanensis* gen. et sp. nov, from the Yixian Formation of China. The debut of *Archaebuda* in the Yixian Formation provides first-hand material for debate on the early evolution of angiosperm flowers and underscores the great diversity of angiosperms in the Yixian Formation.

## 1. Introduction

Angiosperms have more than 300,000 species spread over Earth’s ecosystems, accounting for more than 90% of the species diversity of land plants. They are not only an indispensable component of the ecological background, in which human beings originated and evolved, but they also supply most of the materials necessary for the sustainable development and survival of humans. Despite the importance of angiosperms, their origin and early evolution have been perplexing and mysterious, therefore, these questions have become foci of botanical debates for a long time. Fossil evidence plays an important role in our understanding of the origin and evolution of angiosperms and their flowers. The discovery of the bisexual reproductive organs of Bennettitales [1] became crucial fossil evidence for Arber and Parkin [2], who proposed Magnoliales as the ancestral type in angiosperms, and this school of botany was highly dominant in botany until the 1990s. It is obvious that fossil evidence can lend a hand to botanists in their academic debates. Unfortunately, fossils are not always easy to access, especially those of the more fragile parts of plants, such as flower buds. Early angiosperms, including *Archaefructus* and other pioneer angiosperms [3,4,5,6,7,8,9,10,11,12,13,14,15], have been repeatedly documented from the Yixian Formation of northeastern China. These valuable specimens have supplied important first-hand materials for the debates over angiosperm evolution. However, flower buds have so far been lacking in the Yixian Formation. This lack of information makes many hypotheses on flower evolution open to question. To make our knowledge of early angiosperms comprehensive, we report here a flower bud fossil from the Yixian Formation (Lower Cretaceous). *Archaebuda* is the first flower bud found from the Yixian Formation. The debut of *Archaebuda* in the Formation adds to the already great diversity of angiosperms in the Lower Cretaceous and sheds new light on the evolution of early flowers.

## 2. Materials and Methods

The Yixian Formation of Northeastern China has yielded various fossilized animals and plants [3,4,5,7,8,9,10,11,12,13,14,15,16,17,18,19,20,21,22,23,24,25,26,27,28,29,30,31,32,33,34,35,36], and stratigraphers have intensively studied the formation [37,38,39,40,41,42,43,44,45,46]. Although the age of the formation used to reach up to the Jurassic, recent works converge on a general consensus of approximately 125 Ma (Barremian-Aptian, Lower Cretaceous) [47]. The fossil specimen studied here was collected from the Dawangzhangzi outcrop of the Yixian Formation near Lingyuan, Liaoning, China (Figure 1). The specimen was preserved as a compression/impression embedded in thin-layered siltstone. The specimen is 32 mm long and 8.6 mm wide, preserved on slightly yellowish-grey siltstone (Figure 2a). The whole specimen was imaged using a Nikon D300S digital camera. Morphological details were imaged using a Nikon SMZ1500 stereomicroscope equipped with a Nikon DS-Fi1 digital camera and a Leo 1530 VP scanning electron microscope (SEM) at the Nanjing Institute of Geology and Palaeontology, CAS (NIGPAS). A sketch was drawn from photographic images of the specimen, and all figures are organized using Photoshop 7.0.

## 3. Results

**Genus** *Archaebuda* gen. nov.

**Generic diagnosis**: Flower bud—long-stalked. Stalk—straight, bearing scaly leaves. Scaly leaves—long–triangular and spirally arranged along the stalk. Bud—elongated oval, including two types of foliar appendages. Foliar appendage of type I—smaller, keeled, and round-tipped. Foliar appendage of type II—bigger, keel free, papery, notched at the tip, overlapping each other, and consisting of at least three layers. Stamens and gynoecium—unknown.

**Type species**: *Archaebuda lingyuanensis* gen. et sp. nov.

**Etymology**: *Archae-* Latin word for ancient and *-buda* from the English word *bud*.

**Species***Archaebuda lingyuanensis* sp. nov.

**Species diagnosis:** The same as that of the genus.

**Description:** The specimen includes a stalk and a flower bud on the terminal of the stalk (Figure 2a). The stalk is slender, long, and straight, 15.5 mm long and 1.9 mm wide, bearing scaly leaves (Figure 2a). The scaly leaves are 3–3.5 mm long and 0.9–1.1 mm wide, spirally adnate to the stalk for 85% of its length, only 15% of the distal portion is free from the stalk, slightly crenate-margined, keel-free, long–triangular, with longitudinally oriented epidermal cells (Figure 2a–f and Figure 3a,b). The bud is elliptical in shape, 17 mm long and 9 mm wide (Figure 2a,b). Foliar appendages of type I are at the base of the bud, smaller than the foliar appendages of type II, 2.6 mm long and 3 mm wide, keeled, with a round tip, and longitudinally oriented epidermal cells (Figure 3c,d). Foliar appendages of type II are bigger than the foliar appendage of type I, 4.8–16.8 mm long and 4.1–7.6 mm wide, keel-free, papery, frequently notched at the tip, overlapping each other, consisting of at least three layers, and with longitudinally oriented epidermal cells (Figure 2b,c and Figure 3a,b). 

**Etymology**: *lingyuan*-, for Lingyuan City, Liaoning, China, the fossil locality.

**Holotype**: 20130506025 (Figure 2 and Figure 3). 

**Type locality**: Dawangzhangzi, Lingyuan, Liaoning, China (41°15′ N, 119°15′ E, Figure 1).

**Type horizon and age**: the Yixian Formation, equivalent to the Barremian-Aptian, Lower Cretaceous (approximately 125 Ma). 

**Depository**: the National Orchid Conservation Center of China and the Orchid Conservation & Research Center of Shenzhen, Shenzhen 518114, China.

## 4. Discussions

The scaly leaves on the stalk of *Archaebuda* are adnate to the stalk for most of their length, with triangular pointed tips (Figure 2d–f). Among known gymnosperms, scaly leaves of similar morphology are frequently seen in conifers, but so far have never been seen in Bennettitales, Ginkgoales, and Gnetales [48,49,50,51,52,53,54]. Although cataphylls similar to scaly leaves in *Archaebuda* are seen in the bottom of some cycad cones, these cataphylls have distal pricks that are lacking in the scaly leaves of *Archaebuda* (Figure 2d–f), and numerous peltate shields or tapering segments of “sporophylls” on the surface of cycad cones [55,56,57,58,59] distinguish cycads from *Archaebuda* (Figure 2a–c). The reproductive organs in these gymnosperms (including cycads) usually have either isolated projecting ovules/seeds (Ginkgoales) or radially arranged lateral appendages (Bennettitales and Gnetales) [48,60,61,62] rather than longitudinally-oriented foliar appendages in at least three layers as in *Archaebuda* (Figure 3a,b) and *Magnolia* (Appendix A). Furthermore, all reproductive organs in gymnosperms (fossil and extant) are never known to have more than two layers of papery lateral appendages overlapping each other as in *Archaebuda* (Figure 3a,b). These differences distinguish our fossil from Bennettitales, Ginkgoales, Pentoxylales, Corystopsermales, Peltaspermales, and Gnetales [48,49,50,51,52,53,54]. Therefore, these gymnosperm groups are not further considered. 

Since scaly leaves similar to those of *Archaebuda* are seen in some Coniferales, it is necessary to exclude conifers before further consideration. Reproductive organs play a more important role in plant taxonomy than foliages do. Some conifer cones (especially those of *Abies* and *Picea* in Pinaceae) [52,53,63,64] indeed demonstrate a certain resemblance (in general profile and terminal position) to *Archaebuda*. However, the following differences are obvious enough to distinguish them from *Archaebuda*: *One*, none of the gymnosperms have more than two layers of their lateral appendages overlapping each other, in strong contrast to three layers of overlapping papery foliar appendages seen in Figure 3a,b. This difference alone is enough to distinguish *Archaebuda* from all gymnosperm cones (fossil and extant). *Two*, although both are stalked, the stalk of *Archaebuda* is much more slender and elongated (as seen in tulip and poppy flowers) (Figure 2a) while the stalks are much more stout in most conifer cones [65,66] (except *Amentotaxus*, which has an elongated stalk but its integral aril is distinct from the overlapping papery foliar appendages in *Archaebuda*) [63]. *Three*, a conifer cone usually has multiple lateral appendages helically and vertically arranged around a central axis, and its surface is usually covered with apophyses (except *Calocedrus* and *Juniperus*) [52,53,54,63,64,66] (Appendix A), while the surface of *Archaebuda* is relatively smooth, covered with papery foliar appendages lacking any special features except notched tips (Figure 2b and Figure 3a,b). Indeed, there are exceptional cones in *Calocedrus* and *Juniperus* that have a single bract covering almost the whole length of the cones [64,67], and thus demonstrate more similarity to *Archaebuda* than other conifer cones do. However, the bracts of *Calocedrus* have distal pricks, in contrast to the rounded tip (sometimes notched) of foliar appendages of type II in *Archaebuda*. The bracts in *Juniperus* have no distal pricks, but these bracts are fleshy and of only one layer [67], in strong contrast to at least three layers of papery foliar appendages overlapping each other in *Archaebuda*. Furthermore, pinaceous cones are characterized by their bracts separate from the scales and frequently visible on the cone surface [63,68], which are fully lacking in *Archaebuda* (Figure 2a–c). Some cupressoids (especially Athrotaxoideae, extant and fossil) may appear similar to *Archaebuda* in certain aspects. However, a careful examination can easily exclude this possibility. There are at most two lateral appendages of conifer cones partially overlapping each other, while there are three or more layers of overlapping papery foliar appendages in *Archaebuda* (Figure 3a,b). The latter case is routine and typical in flower buds of living angiosperms, for example, in *Magnolia* (Appendix A) and many other angiosperms. *Four*, the foliar appendages of type I and II in *Archaebuda* are of distinct sizes and morphologies (Figure 3a–d) while lateral appendages (bract–scale–seed complexes) in conifer cones are either uniform or of a gradually transitional appearance (Appendix A) [52,53,54,63,64]. All these differences culminate in the conclusion that *Archaebuda* is not a conifer cone, and that it represents a flower bud in the Early Cretaceous.

Due to intensive palaeobotanical studies [3,4,5,6,7,8,9,10,11,12,13,14,15], the great diversity of angiosperms in the Yixian Formation has been repeatedly underscored as various reproductive organs of angiosperms have been reported [3,4,5,69]. According to common sense in palaeontology, fragile parts of organisms have less potential to be preserved as fossils. This understanding has been a cornerstone of paleontological practice over past centuries. An implication of this is that fragile parts of plants, such as flowers, flower buds, and cytoplasm, are not expected in the fossil world. However, as palaeontology has developed, exceptions to this rule have frequently occurred. For example, reproductive organs of angiosperms (flowers) have been documented in the Yixian Formation [3,4,5,6,9,69], cellular ultrastructures and chloroplasts have been seen in Eocene *Metasequoia* [70], and even exocytosis snapshots have been observed in a Miocene conifer [71,72]. These exceptions remind us that, theoretically, fragile parts of plants can also be preserved in fossils. However, the reality is that fragile flower buds have been a rarity in the fossil record [73], especially in the Yixian Formation, which is famous worldwide for its diversity of angiosperms [3,4,5,6,7,8,9,10,11,12,13,14,15]. 

The numerous reproductive organs of angiosperms reported from the Yixian Formation [3,4,5,6,7,8,9,10,11,12,13,14,15] (except for *Archaefructus*) differ from each other morphologically, underscoring the great diversity of angiosperms in the Yixian Formation. Despite such a great diversity of angiosperms, hitherto there was no fossil record of flower buds in the Yixian Formation. None of the previously reported angiosperms in the formation has perianth, as *Arachaefructus* [3,4,5,8] and *Sinocarpus* [11,12] indicate. The present report of *Archaebuda* implies that flowers with perianth (as in most angiosperm flowers) occurred 125 Ma ago. This discovery is helpful to decouple the evolutionary history of flowers from that of angiosperms. The repeated popping up of new angiosperms in the Yixian Formation implies that angiosperms in the Early Cretaceous are still under-studied. 

A Jurassic peer of *Archaebuda*, *Florigerminis* [73], seems to favor an earlier origin of angiosperms. However, both fossils are not permineralized and thus prevent more detailed discussion. Perhaps future application of Micro-CT or other new technology will help to reveal more internal and crucial information in better preserved fossils that will help palaeobotanists to make robust conclusions. 

## 5. Conclusions

Despite various angiosperms reported from the Yixian Formation, until now, none of them has been a flower bud. *Archaebuda* reported here represents the debut of flower buds in the Early Cretaceous, indicating that flowers with a perianth did exist 125 Ma ago. This is a key contribution to the study of flower history.

## Figures and Tables

**Figure 1 biology-11-01598-f001:**
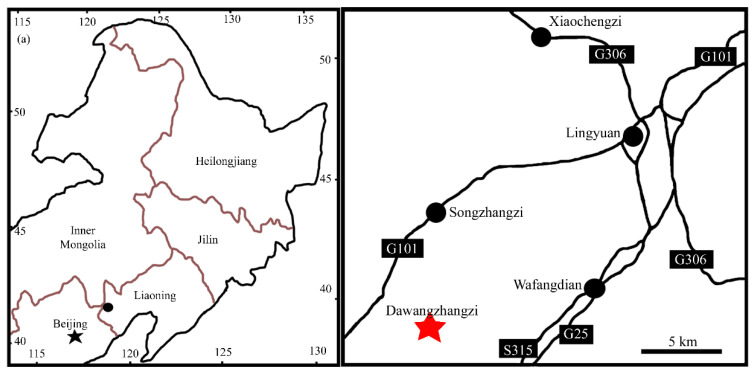
Geographical information of the fossil locality for *Archaebuda* gen. nov, Dawangzhangzi, Lingyuan, Liaoning, China. Reproduced from Han et al. [14], with permission and courtesy of Acta Geologica Sinica (English edition). **Left**. Fossil locality (black dot) in northeastern China. **Right**. Detailed position of fossil locality (red star) in a suburb of the city Lingyuan, Liaoning.

**Figure 2 biology-11-01598-f002:**
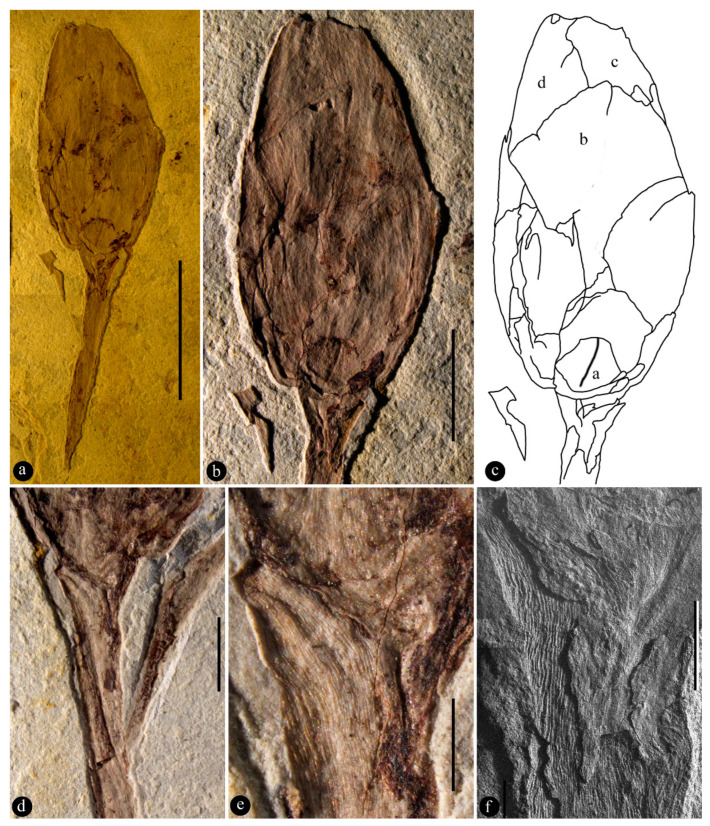
The holotype of *Archaebuda* gen. nov. and its details. (**a**) The general view of the specimen, including a stalk and a flower bud. Bar = 10 mm. (**b**) Detailed view of the bud composed of various overlapping patches. Bar = 5 mm. (**c**) Sketch of the bud in Figure 2b. Parts marked by letters (b, c, d) are shown in Figure 3a. (**d**) Basal portion of the flower bud. Bar = 1 mm. (**e**) Detailed view of the scaly leaf with cellular details on the pedicel. Bar = 1 mm. (**f**) SEM view of the portion in Figure 2e, showing the longitudinal arrangement of the cells. Bar = 1 mm.

**Figure 3 biology-11-01598-f003:**
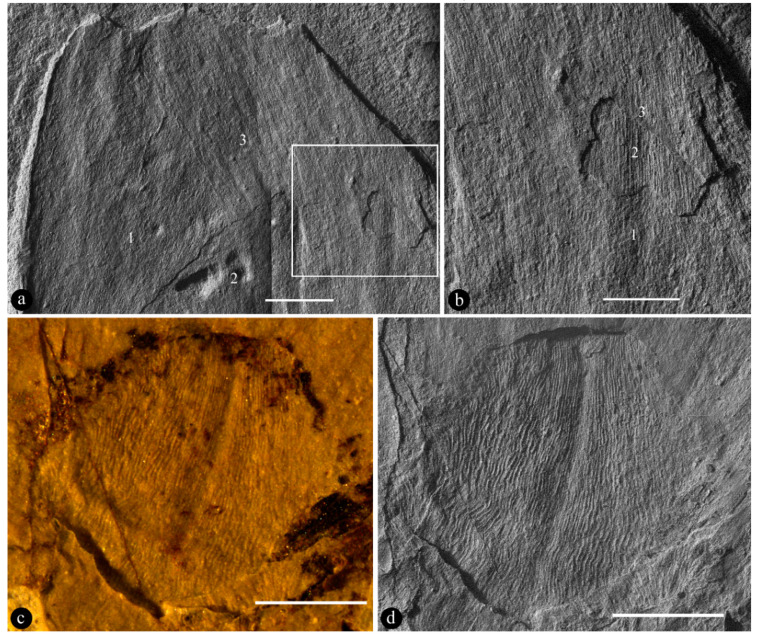
Details of *Archaebuda* gen. nov under SEM, except Figure 3c. (**a**) Detailed view of the tip of the flower bud, showing at least three overlapping papery foliar appendages of type II (1,2, 3, corresponding to d, c, b in Figure 2c) with differently oriented textures. Bar = 1 mm. (**b**) Detailed view of the rectangle in Figure 3a, showing three layers of overlapping papery foliar appendages of type II (1, 2, 3). SEM. Bar = 0.5 mm. (**c**) Stereomicroscopic view of a keeled foliar appendage of type I at the base of the flower bud. Bar = 1 mm. (**d**) SEM image of the foliar appendage of type I shown in Figure 3c, showing surface texture. Bar = 1 mm.

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
