# Peer review of "A Flower Bud from the Lower Cretaceous of China"

_biology, 2022, doi:10.3390/biology11111598_

Round 1
Reviewer 1 Report (Previous Reviewer 4)
I found the manuscript too short and vague in discussion. Even the fossil seems to be wonderful preserved with vegetative and probable reproductive characters in connection, authors do not reach an important discussion in order to provide disambiguation among different plant taxa.
As starting I begin my reviewing with the open mind to obtain from the text, several anatomical and morphological characters that certificate angiosperm assignation, but I could find no single one. Furthermore, I find authors reach to an angiosperm assignation trying to discard other plant groups and omitting most of the reproductive diversity that gymnosperms provide in both: extant and fossil taxa. Also, in same way authors make a very lightly discussion about general characters on higher rank taxa without a proper discussion about the diversity gymnosperms and Coniferales possess about the scale, bract, ovule and ovuliferous complexes. The use of terms “typical cone” or “typical flower” are odd when should be discussed anatomical and morphological characters used as synapomorphies of the groups.
Several of my previous indications were not done.
I wish authors provide more descriptions over those leaves (% of attached part, shape of the free part, margin and tip shape, presence of a kill, etc..).
I agree with authors the scales/”tepals” are imbricated and disposed one over the other. That is a character mathematically described by the Fibonacci Sequence that is used in plant taxonomy for description and identification among plant groups. I wish to see a proper discussion about the interpretation of this cone/bud/flower with comparisons among major plant groups and synapomorphyes of clades (Angiosperms and gymnosperms).
To me, and base on the figures, the bracts on the “bud” surface are still in a helical arranged forming not vertical orthosis. Also elongation of the epidermal cells and the well developed kill and perpendicular epidermal cells are telling that the bract is in almost end of their organ expansion, so really is not a slender delicate organ in developing. The position of these bracts, shapes and epidermis remind me very much to the cones in Athrotaxoideae (extant and fossils species).
Authors are explaining that they doesn´t find obvious apophyses on the scales as in some conifers, but do not explore the total diversity present in gymnosperms or Coniferales, furthermore do not give any detail about the fossil they are studying.
At the point of the description and discussion provided in the manuscript, seems obvious to me (and in agreement with the authors), that the fossil doesn’t´match with any extant conifer or Gymnosperm genus. But a more deep discussion is needed in order to discard all other reproductive structures from fossil gymnosperms. a more deep discussion is needed in order to assign the fossil to an angiosperm, providing almost a single character that qualify as an angiosperm. Furthermore, a deep discussion is needed in order to classify the fossil as a reproductive structure due authors doesn´t provide any reproductive character other than “enwrapping” (as they called) bracts. Bracts are modified leaves and are suitable to be founded even in vegetative winter leafy buds or at the apex of dwarf shoots.
I found several circular thinking, erroneous asseverations due to too much generalization (lacking a complete discussions) and absence of strong characters in order to accept any assignation to the plant material.
I marked the document in several spots but I quit due already explained my comments in previous revision. Figure moved to Supplementary (S1) still is not useful for the deep anatomical and morphological comparison the fossil request in order to be assigned to a plant group. In other word, need to be replaced by more conclusive cone or bracts resembling the fossil.
Base on above, and the nice preservation of the material and importance of the age and geography, I still recommend it to “Reconsider after major revision”.

Author Response
Thanks for your kind help. Please see detailed information in attached Word file.

Reviewer 2 Report (Previous Reviewer 3)
This is an interesting manuscript describing a plant fossil that can be a flower bud. However, its preservation is insufficient to state the fact it is an angiosperm.
In the diagnosis the author uses the term flower bud, but this is inadequate. It could be other type of structure.
This structure can represent e.g. a pollen cone of some gymnosperm (pollen in situ is necessary to exclude this).
Another disadvantage is the lack of other type material except of the holotype.
Some minor items:
Line 98: Instead of “Stratigraphic horizon“ I recommend “Type horizon and age“.
Lines 184-207 can be excluded because they are discussing diversity of angiosperms in Yixian Formation, but the presently described fossil cannot add much to this discussion being porel preserved and ambiguous.
Author Response
Thanks for your kind help. Please see detailed information in attached Word file.

Round 2
Reviewer 1 Report (Previous Reviewer 4)
I checked the revised version again. Redaction was performed and several of my comments was proper adresed. Seems ok to me. I saw the the reply from authors to my last comments. I can acept all of them. But still is a shame that they couldn´t provide any strong character in favor for to the angiosperm identification, other than discard several gimnosperms. Angiosperm characters that I wish to observe in that kind (and pictures) of a fossil are epidermal cell shape, stomata shape and type, trichomes, epidermal cells at the "petal" margin and at the notched apex.Reviewer 2 Report (Previous Reviewer 3)
I am sorry I do not have time to deal with rised questions. After editing of the manuscript it seems acceptable in present form.
Author Response
Open Review
( ) I would not like to sign my review report
(x) I would like to sign my review report
English language and style
( ) Extensive editing of English language and style required
( ) Moderate English changes required
(x) English language and style are fine/minor spell check required
( ) I don't feel qualified to judge about the English language and style
Yes |
Can be improved |
Must be improved |
Not applicable |
|
Does the introduction provide sufficient background and include all relevant references? |
(x) |
( ) |
( ) |
( ) |
Are all the cited references relevant to the research? |
(x) |
( ) |
( ) |
( ) |
Is the research design appropriate? |
(x) |
( ) |
( ) |
( ) |
Are the methods adequately described? |
(x) |
( ) |
( ) |
( ) |
Are the results clearly presented? |
(x) |
( ) |
( ) |
( ) |
Are the conclusions supported by the results? |
(x) |
( ) |
( ) |
( ) |
Comments and Suggestions for Authors
I am sorry I do not have time to deal with rised questions. After editing of the manuscript it seems acceptable in present form. Thanks for your accepting our manuscript.
Submission Date
26 September 2022
This manuscript is a resubmission of an earlier submission. The following is a list of the peer review reports and author responses from that submission.
Round 1
Reviewer 1 Report
The ms is excellently written and represents quite an interesting finding. I did not notice any severe mistakes that should be fixed so that I can recommend this ms for publication without restrictions.
----------------------------
-What is the main question addressed by the research? Is it relevant and interesting?
--> The main question of the ms is a description of the new genus and species based on a flower bud from the Yixian Formation. The authors also provided detailed arguments on how they decided that it is a bud, but not a cone (i.e., arrangement of lateral appendages, their shape, and overlapping pattern). As for me, such a description (just as most of the paleobotanical reports) is important and quite interesting because it helps to fill the gap in the evolution of angiosperms and to understand the morphology and phylogeny of these plants.
-How original is the topic? What does it add to the subject area compared with other published material?
-->The ms is totally original and provides a description of a new fossil bud. Such genus and species were not reported before. This report complement already existing published materials on Cretaceous paleobotany and is important for further reconstructions of floral phylogeny in angiosperms and other evolutional studies.
-Are the experimental methods and design reasonable? Is there any information that needs to be improved or added?
--> Yes, the menthods and material are well described and fully acceptable. I see no necessity to provide any additional information. However, it would be great if the authors would indicate where exactly the holotype 20130506025 is deposited.
-Is the paper well written? Is the text clear and easy to read?
-->Yes, the ms is well-written and readable. This I already indicated in the radio-button review form that you provided.
-Are the conclusions consistent with the evidence and arguments presented? Do they address the main question posed?
--> The same here, these questions repeate the radio-button form that you provided and are already responded by me. Conclusions are brief but fully correspond to provided materials. This ms, in general, is just a report of a new fossil finding so there is nothing super-hard to review or something extra-speccila to discuss. It is simple, concise, and well-prepared, so I see no any restriction on accepting it.
Reviewer 2 Report
The manuscript presents a single specimen of a bud with very little morphological and anatomical detail. This is insufficient to warrant publication of such result. This specimen should be presented in a more general view of similar structures from the same formation, with much less overstatement about its supposed importance for the field of angiosperm paleobotany.
Reviewer 3 Report
This is an interesting manuscript describing a plant fossil that can be a flower bud. Its preservation is however insufficient.
My major concern is the diagnosis. The use of the term flower is here inadequate. It could be other type of structure. (I cannot accept the authors statement excluding Bennettitales, Ginkgoales, Cycadales, and Gnetales.) The diagnosis should not contain terms as sepals and tepals that pre-maturely interpret the structure.
Another disadvantage is the lack of other type material except of the holotype.
Because the fossil is an impression/adpression (the author states “compression/impression“ without further explanation) no pollen can be found there and therefore its interpretation is difficult. Due to lack of any suitable character its interpretation as an angiosperm flower is just a speculation.
Line 58: If the preservation is “compression/impression“, the author should try pollen in situ, that would bring more light to this puzzling fossil.
Line 108: There are structures similar to that described e.g. in Ginkgoales.
Reviewer 4 Report
Revision
Manuscript ID: biology-1847552 Type of manuscript: Communication
A Typical Flower Bud from the Lower Cretaceous of China
I found the manuscript too short and vague in discussion. Even the fossil seems to be wonderful preserved with vegetative and reproductive characters in connection, authors do not reach important discussion that provide disambiguation among different plant taxa.
As starting I begin my reviewing with the open mind to obtain from the text, several anatomical and morphological characters that certificate angiosperm assignation, but I could find no single one. Furthermore, I find authors reach to an angiosperm assignation discarding other plant groups and omitting most of the reproductive diversity that gymnosperms provide in both: extant and fossil taxa. For instance, the only place where they discuss probable gymnosperm (other than conifer) assignation is constrained to two lines : 108 and 109.
Even more, base on that the following discussions are only compared to extant conifers (a group very derivate in gymnosperms). Furthermore, the only lines were they provide any comparison to Coniferae is in lines 130-135 and only using very derivate Pinaceae genera (Abies, Picea and Cedrus).
Characters that I can recognize base on pictures:
I agree with authors there are scaly leaves on the stem/stalk that do not need to indicate the sole gymnosperm assignation. But I wish authors provide more descriprions over those leaves (% of attached part, shape of the free part, margin and tip shape, presence of a kill, etc..).
I agree with authors the scales/”tepals” are imbricated and disposed one over the other. That is a character mathematically described by the Fibonacci Sequence that is used in plant taxonomy for description and identification among plant groups. I wish to see a proper discussion about the interpretation of this cone/bud/flower with comparisons among major plant groups.
To me, and base on the figures, the bracts on the “bud” surface are still in a helical arranged forming not vertical orthosis. Also elongation of the epidermal cells and the well developed kill and perpendicular epidermal cells are telling that the bract is in almost end of their organ expansion, so really is not a slender delicate organ in developing. The position of these bracts, shapes and epidermis (and enwrapping scales opposite to manuscript sayd in line 126) remind me very much to the cones in Athrotaxoideae (extant and fossils species).
Authors are explaining that they doesn´t find obvious apophyses on the scales as in some conifers, but do not explore the total diversity present in gymnosperms or Coniferales (line 130), furthermore do not give any about the fossil they are studying.
I found several circular thinking, erroneous asseverations due to too much generalization (lacking of complete discussions) and absence of strong characters in order to accept any assignation to the plant material.
I marked the document in several spots. Due to the needed of reconsider the assignation and/or further develop the descriptions , several paragraphs at the end should be completely modified.
Base on above, and the nice preservation of the material and importance of the age and geography, I recommend to “Reconsider after major revision”.
